# Measurements of natural airflow within a Stevenson screen, and its influence on air temperature and humidity records

Stephen D. Burt[1]

[1] Department of Meteorology, University of Reading, RG6 6ET, UK

*Correspondence to*: Stephen Burt (s.d.burt@reading.ac.uk) ORCID ID 0000-0002-5125-6546

**Abstract**

Climate science depends upon accurate measurements of air temperature and humidity, the majority of which are still derived from sensors exposed within passively-ventilated louvred Stevenson-type thermometer screens. It is well-documented that, under certain circumstances, air temperatures measured within such screens can differ significantly from 'true' air temperatures measured by other methods, such as aspirated sensors. Passively-ventilated screens depend upon wind motion to provide ventilation within the screen, and thus airflow over the sensors contained therein. Consequently, instances of anomalous temperatures occur most often during light winds when airflow through the screen is weakest, particularly when in combination with strong or low-angle incident solar radiation. Adequate ventilation is essential for reliable and consistent measurements of both air temperature and humidity, yet very few systematic comparisons to quantify relationships between external wind speed and airflow within a thermometer screen have been made. This paper addresses that gap by summarising the results of a three month field experiment in which airflow within a UK-standard Stevenson screen was measured using a sensitive sonic anemometer, and comparisons made with simultaneous wind speed and direction records from the same site. The mean in-screen ventilation rate was found to be 0.2 m s$^{-1}$ (median 0.18 m s$^{-1}$), well below the 1 m s$^{-1}$ minimum assumed in meteorological and design standard references, and only about 7% of the scalar mean wind speed at 10 m. The implications of low in-screen ventilation on the uncertainty of air temperature and humidity measurements from Stevenson-type thermometer screens are discussed, particularly those due to the differing response times of dry- and wet-bulb temperature sensors, and ambiguity in the value of the psychrometric coefficient.

## 1. Background and motivation

Accurate measurements of air temperature and humidity require the sensors to be protected from direct or reflected solar and terrestrial radiation and precipitation. The most common exposure for such instruments remains that within a passively-ventilated thermometer screen or radiation shelter, of which there are many different varieties and patterns in use worldwide: many can be broadly classed as Stevenson-type thermometer screens, otherwise known as Cotton Region Shelters in the US (Burt, 2012, Chapter 5). These typically comprise a double-louvred enclosure, traditionally of wood painted gloss white, but

increasingly of UV-resistant glossy white plastic, with double roof and base. The drawback of this type of thermometer screen (one shared by the smaller multiplate radiation shields typically used in automatic weather stations) is that the double-louvred construction, whilst effective at reducing radiation exchange with its surroundings, also acts as a very significant barrier to natural ventilation through the body of the screen. Such reduction in airflow may result in significant and persistent departures in air temperature and humidity from 'true' conditions, including excess warming of the screen interior and increases in sensor response time and extended screen lag times, especially when conditions are also changing rapidly. In addition, variations in the psychrometric coefficient at low airflow increase uncertainty in the determination of humidity parameters. Two or more of these factors often occur simultaneously and may persist for considerable periods of time, both in daylight (strong sunshine, light winds) or at night (when wind speeds tend to be lower). Implications and consequences, with examples, are considered in more detail subsequently.

It is therefore surprising that few measurements of ventilation speed have been attempted within Stevenson screens, perhaps because instruments combining unidirectional sensitivity to very low air flow speeds ($< 0.1$ m s$^{-1}$) are a relatively recent innovation. Limited investigations in Poland by Swioklo (1954) suggested that wind speeds inside screens amounted to about 10% of external wind speeds, while Folland (1977) suggested 15% of 10 m wind speeds, based upon 19 data points. A similar investigation by Bultot and Dupriez (1971), at Uccle in Belgium, showed that in a large screen a ventilation rate of 1 m s$^{-1}$ was reached only very exceptionally, and that more often it was between 0.2 and 0.6 m s$^{-1}$. More recently, Lin et al (2001) conducted laboratory and field trials to assess the airflow efficiency of various types of thermometer screen, including Cotton Region Shelter models as well as multiplate radiation shields. Dobre et al (2018) attempted to model internal ventilation rates within a Stevenson screen, and suggested that airflow $< 0.35$ m s$^{-1}$ was most typical. In contrast, ISO 17714 (International Organization for Standardization, 2007) simply assumes a ventilation rate of 1 m s$^{-1}$ within modern Stevenson-type screens.

This field experiment did not attempt a detailed investigation into the variation of airflow *within* the screen. This could probably be assessed using an array of small, high-resolution airflow sensors within the screen to obtain point measurements with which to build a high-resolution computational fluid dynamics (CFD) model, although to minimise external influences, the experiment would be best undertaken within a wind tunnel. A very large number of different measurements would be required, varying both direction and speed of the airflow over the screen. Some work regarding a CFD approach to model within-screen airflow has been attempted (Dobre et al, 2018).

## 2. Experimental arrangements

A field experiment was chosen instead of laboratory wind tunnel tests at the outset, because the objective of the research was to quantify actual ventilation rates within a typical modern thermometer screen over a representative period under normal outdoor exposure conditions. Assessing airflow around and within a screen mounted within wind tunnel would certainly permit greater control of ambient airflow, but at the risk of imperfectly representing the range of conditions within an outdoor environment, including of course solar radiation and precipitation.

*Screen airflow measurements* A sensitive Gill Windsonic anemometer was installed within a standard plastic-and-aluminium Metspec Stevenson screen within the meteorological enclosure of the University of Reading Atmospheric Observatory (51.441°N, 0.938°W, 66 m AMSL), itself located in an open position in a parkland campus. The screen was free of nearby external obstructions on all sides (Figure 1). Within the screen, the Windsonic unit was mounted with its measurement aperture horizontal, oriented accurately towards true north, and close to the centre of the screen interior at about 1.25 m above ground level in a position representative of the typical location of temperature and humidity sensors within such a screen within the United Kingdom (Figure 2). The instrument was secured in place within the screen by a laboratory retort stand, itself fixed in place by cable ties to prevent any movement of the sensor during the experiment. The screen was otherwise empty to avoid any obstructions to airflow within the screen from other equipment and fittings, and was kept padlocked to prevent the screen door being opened during the experimental period. Data were logged by a Campbell Scientific CR1000 logger housed externally to the screen; the sensor was sampled at 1 Hz and logged at 1 min, 5 min and hourly intervals. Logged samples included average, minimum and maximum speeds, and vector mean directions. For most of the analysis, hourly means were found sufficient, although 1 min and 5 min records were available and were examined where additional granularity was beneficial.

*Calibration and uncertainty* The Windsonic unit was less than 6 months old at the outset of the measurement programme. The manufacturer's calibration uncertainty is within ±2% at 12 m s$^{-1}$, with a resolution of 0.01 m s$^{-1}$; the manufacturer's documentation attests that 'the unit is designed not to require re-calibration within its lifetime'. A still-air test as set out by the manufacturer was carried out before and after the experiment to verify that no zero-offset applied to the output, and both tests were negative. Following the in-screen experiment, the unit was exposed for six months at 3 m above ground level alongside another Vector Instruments A100L cup anemometer of known manufacturer calibration. Above 1 m s$^{-1}$ wind speed, and over a wide range of speeds, both instruments agreed within 2% as expected. However, discontinuities resulting from the starting/stopping speed of the cup anemometer at about 0.4-0.5 m s$^{-1}$ rendered impractical a similar comparison at speeds below 1 m s$^{-1}$, as a result of which there remains some uncertainty as to whether the manufacturer's ±2% specification is valid at the lower speed ranges found to be typical of in-screen ventilation rates. The possible impact of this uncertainty upon derived conclusions is considered in more detail in the Results section, but even the assumption of a much greater uncertainty (+20%) on logged values at low speeds does not significantly affect the general conclusions of the work.

The anemometer was installed on 13 February 2020, and data were logged continuously until the unit was removed on 27 May 2020, except for three days record being lost 11-14 April. The latter was a result of the closure of the university campus due to coronavirus emergency measures: unfortunately data from the logger could not be retrieved before being overwritten. Access to the Observatory and logger was next possible on 7 May, when all data since 14 April were successfully downloaded. In all, records from 145 433 minutes (2423 hours, 101 days) were available for this analysis.

*External wind speed and direction measurements* Routine measurements of wind speed at 2 m and 10 m above ground, together with wind direction at 10 m, are also made within the Observatory enclosure using Vector Instruments cup anemometers calibrated in accordance with manufacturer recommendations with an expected uncertainty of ± 0.1 m s$^{-1}$. These are sampled

and logged every second using a Campbell Scientific CR9000X logger (along with numerous other sensors and research instruments within the meteorological enclosure). Hourly means (resolution 0.1 m s$^{-1}$) were used in this analysis, with shorter time periods available for examination as required.

*Rationale for the choice of differing instruments* The classical approach to an experiment of this type would suggest identical instruments be chosen for both interior and exterior measurements, in order to compare like with like. However, for this experiment the Gill Windsonic sensor was consciously chosen for in-screen airflow measurements because previous pilot tests had demonstrated its ability to provide reliable measurements of wind speed at very low ventilation rates, typically well below the expected starting or stopping speeds of conventional cup anemometers. For external measurements, records from existing conventional cup anemometers at standard heights (2 m and 10 m above ground level) were preferred precisely because of the widespread availability of similar records within meteorological data series, thus enabling the results from this experiment to be widely and directly applicable to conventional wind records made elsewhere using similar instrumentation. The external anemometers were also available and well-maintained.

## 3.    Results and analysis

The principal timescale used in this analysis is that of hourly averages; 1 minute and 5 minute analyses showed greater scatter, as expected, but were otherwise almost identical in pattern to hourly evaluations. During the experimental period, the distribution of wind direction was bimodal, the frequency of winds from between south-west and west approximately equalling that from winds between north-east and east (see also 'Wind directions', below). Hourly mean speeds at 10 m ranged from zero to 9.6 m s$^{-1}$; the maximum 3 s wind gust at 10 m was 21 m s$^{-1}$, on 15 February. Aside from the abnormally high frequency of north-easterly winds during the second half of the period, wind conditions were climatologically representative of this mid-latitude inland site—the mean wind speed at 10 m during the experimental period was 2.8 m s$^{-1}$, a little greater than the average for February to May (2.4 m s$^{-1}$ over the preceding 5 year period) but similar to observed mean annual wind speeds.

### 3.1 Ventilation speeds within the screen

For the analysis period (13 February to 27 May 2020, excluding 11-14 April), the entire dataset consisting of 2423 hourly means of wind direction and speed at 2 m and 10 m was used to compare external wind speed and direction with those measured using the sonic anemometer within the screen. Figure 3 shows hourly means of the in-screen ventilation speed $U_{screen}$ plotted against the simultaneous external hourly scalar mean wind speed at 10 m ($U_{10}$). A very similar pattern was apparent for winds at 2 m ($U_2$, Figure 4), the scale differing only in reflecting the reduction in wind speed at this height. Two subsets of the data were examined in detail, as explained below.

### 3.1.1 External wind speeds above about 1 m s$^{-1}$

At exterior wind speeds above 1 m s$^{-1}$, referencing either $U_2$ or $U_{10}$ as appropriate, screen ventilation $U_{screen}$ was close to a linear function of $U_{10}$ (Figure 3) and $U_2$ (Figure 4). To a near approximation, and when considering hourly means, $U_{screen}$ averaged just 7% of $U_{10}$ (Figure 5) and 10% of $U_2$ (Figure 6) in this subset, the ratio decreasing slightly with increasing wind speed at both levels (Table 1). Thus, to a reasonable approximation for external wind speeds $\geq 1$ m s$^{-1}$, values of $U_{screen}$ for external wind measurements made at 2 m above ground are given in equation 1, and similarly for measurements at 10 m in

equation 2:

$$(1) \quad U_{screen} \approx U_2 * 0.10 \text{ when } U_2 > 1 \text{ m s}^{-1}$$

*50% observations within $U_2 * 0.10 \pm 0.01$, 90% observations between $U2 * 0.08$ and $U2 * 0.13$*

$$(2) \quad U_{screen} \approx U_{10} * 0.07 \text{ when } U_{10} > 1 \text{ m s}^{-1}$$

*50% observations within $U_{10} * 0.07 \pm 0.01$, 90% observations between $U_{10} * 0.05$ and $U_{10} * 0.11$*

### 3.1.2 Light winds (external wind speeds below 1 m s$^{-1}$)

In light winds, $U_{screen}$ occasionally exceeded $U_2$ (20 hours in all, around 1% of analysis period; to avoid undue compression of the y-axis these points have been omitted from Figure 6). However, this unlikely outcome is simply explained: the sonic

anemometer used for this experiment is capable of resolving wind speeds as low as 0.01 m s$^{-1}$, whereas the cup anemometers used for the exterior wind records have a stopping speed of 0.4-0.5 m s$^{-1}$. At low exterior wind speeds, therefore, the interior to exterior ratio becomes artificially high; in reality, the ratio of $U_{screen}$ to $U_2$ and $U_{10}$ for wind speeds < 1 m s$^{-1}$ is probably little different to that for winds $\geq 1$ m s$^{-1}$.

However, the conclusions set out in the paper are insensitive to even fairly large uncertainties in the sensor's low speed

calibration, for even if the Windsonic unit's calibration at 0.2 m s$^{-1}$ was in error by +20%, or ten times the manufacturer's specification, this would result in only a minor change in the ratio of interior to exterior wind speed (from 10% to 12% for $U_2$, and similarly from 7% to 9% for $U_{10}$). While it is possible that more uncertainty attaches to the lowest speeds, this is largely irrelevant to the outcome, because the typical 0.4-0.5 m s$^{-1}$ stopping speed of the external $U_2$ and $U_{10}$ Vector anemometers meant that reliable comparison ratios could not be accurately obtained below these levels.

### 3.2 Wind direction

Although the direction of airflow within the screen is of lesser importance than its speed, comparisons were made between external hourly vector mean wind directions at 10 m and those measured within the screen by the sensitive sonic anemometer. Close correspondence with exterior wind direction was not expected, but results revealed that airflow within the screen was more complex than anticipated.

### 3.2.1 Wind directions at 10 m

During the experimental period, the distribution of surface winds was almost bimodal, with winds from between south-west and west and those from between north-east and east occurring with approximately equal frequency. Winds from a westerly quarter dominated the first half of the experimental period, and those from the north or east the second half. The wind rose in Figure 7a shows the frequency, by 10° sectors and within six wind speed bins, of hourly vector mean wind directions at 10 m above ground during the experimental period; calms (here taken as hourly scalar mean wind speeds at 10 m below 0.25 m s$^{-1}$) amounted to 2.3%. The outer scale ring represents 2% of all observations.

### 3.2.2 In-screen airflow directions

Figure 7b shows the distribution of hourly vector mean wind directions within the screen over the same period, and in the same format, as the external wind directions in Figure 7a. To enable comparison, the scale on Figure 7b has been expanded such that each speed division is one-tenth of the equivalent exterior speed. Using this classification, 'calm' ($< 0.025$ m s$^{-1}$) represents 10.6% of all events (using the same maximum threshold as exterior wind directions, i.e. 0.25 m s$^{-1}$, the figure would be 68.3%). On this plot, the outer scale ring represents 10% of all 2423 hourly records.

## 4. Discussion

### 4.1 Wind speed

It was surprising to discover that airflow within the screen is very much less than has been conventionally assumed. ISO 17714 (International Organization for Standardization, 2007), for example, assumes a ventilation rate of 1 m s$^{-1}$ within modern passively-ventilated Stevenson-type screens, whereas the results presented here demonstrate that the mean ventilation within the screen during the experimental period was only 0.2 m s$^{-1}$ at this fairly typical well-exposed mid-latitude inland site. In-screen airflow reached 1 m s$^{-1}$ only when the external 10 m wind speed was close to 10 m s$^{-1}$. Further, using 1 minute mean data, an average of 1 m s$^{-1}$ or more was attained for just 17 minutes (0.01%) during the 101 day experimental period. Figure 8 shows the distribution of the 2423 hourly means of U$_{screen}$. Perhaps surprisingly, only 110 minutes averaged 0.00 m s$^{-1}$ airspeed within the screen, less than 0.1% of the dataset, while the lowest hourly mean was just above 0.01 m s$^{-1}$.

Whilst it would be unrealistic to assume that the relationships found in this experiment apply rigidly to all Stevenson-type thermometer screens in any climate, it is clear that an automatic assumption of 1 m s$^{-1}$ internal airflow in such screens is difficult to justify, except perhaps at especially exposed sites where 10 m wind speeds frequently exceed 10 m s$^{-1}$. The low mean airflow rates shown by this experiment have considerable implications for air temperature and humidity measurements, and these are discussed subsequently.

## 4.2 Wind direction

There is a striking difference between 'external' and 'internal' wind roses (Figure 7a, 7b respectively). There is a suggestion here of preferential flow orientations within the screen structure itself, but a more detailed investigation would require additional sensors within the screen to provide a three-dimensional capability. Such an experiment would certainly benefit from a more controlled environment, such as a wind tunnel. However, this experiment found no evidence that the pillars of the screen structure provided any greater obstruction to wind flow from those directions (north-east, south-east etc) than winds orthogonal to one of the screen faces. Despite the low mean speeds, the airflow within the screen appeared to be highly turbulent in nature.

## 4.3 Implications resulting from low screen ventilation

The results presented have important, albeit unavoidable, implications for all measurements of air temperature and humidity taken within Stevenson-type thermometer screens where external wind speeds are typically less than 10 m s$^{-1}$. There are four main effects, and it is helpful to think of these as falling into two main categories, with combinations more likely than not. The first category is where effects are primarily due to low wind speeds per se, including excess warming of the screen structure and lengthening of the screen lag time; the second relates to increasing response times of sensors within the screen and changes in the psychrometric coefficient due to decreasing airflow, both being of course themselves a consequence of low external wind speeds. Each is considered briefly in turn.

### 4.3.1 Excess warming of the screen interior

It is well known that certain combinations of light winds with strong solar radiation, or low-level direct sunshine on the screen exterior, can result in excess warming of the screen exterior, which by radiative exchange warms the screen interior. This tendency has been known for well over a century (Gaster 1882, Aitken 1884) and has been reported in numerous screen comparison trials since (see, for example, Sparks 1972, Andersson and Mattison 1991, Clark et al 2014 and references therein, and Harrison and Burt 2021). Such previous and ongoing work indicate that the magnitude of the warming is not infrequently 0.5 K and occasionally amounts to 2-3 K. The primary cause of the positive departure from 'true' air temperatures is the reduced effectiveness as a result of low wind speeds of turbulent advective cooling of a screen structure heated by solar radiation. Limited in-screen airflow, itself obviously another consequence of low external wind speeds, then results in further reductions in turbulent advective heat transport within the body of the screen, as a result of which screen temperatures rise above 'true' air temperature. Such warming persists for as long as the causative conditions (incident solar radiation and/or light winds) remain in place.

### 4.3.2 Lengthening of the screen lag time

Any artificial structure housing thermometers itself has a response time. The importance of ventilation rate around and within thermometer screens was examined by Harrison (2010, 2011), who found that low levels of natural ventilation led to long (5-20 minute) lags in temperatures measured within a Stevenson screen. These effects were found to be both more common and more pronounced at night, when wind speeds are usually lower than during daytime, and have a proportionally greater effect on minimum rather than maximum temperatures (Harrison and Burt 2020).

### 220 4.3.3 Increased response times of sensors within the screen

Burt and de Podesta (2020) examined the response times of a selection of typical meteorological platinum resistance thermometers (PRT), and found their 63% response time $\tau_{63}$ (in seconds) depended primarily on sensor diameter $d$ (mm) and ventilation speed $v$ (m s$^{-1}$). $\tau_{63}$ could be approximately expressed as equation 3 (equation 11 in Burt and de Podesta 2020)

$$(3)\ \tau_{63} \approx 5.6\ \frac{d^{3/2}}{v^{1/2}}$$

Assuming this relationship, the first row of Table 2 presents calculated $\tau_{63}$ response times for a nominal 'bare' PRT of 3 mm diameter at various ventilation speeds, namely 0.2 m s$^{-1}$ (representing the average in-screen airflow found during this experiment), 1.0 m s$^{-1}$ (in-screen airflow assumed by ISO 17714, but found to occur during just 0.01% of the entire experimental period), and 5.0 m s$^{-1}$ (the airflow typical of an aspirated sensor). A 'bare' PRT is one without sleeving around

230 the outer steel sheath, and is typical of those used in meteorological screens as a dry-bulb thermometer (Tdry) to measure 'air temperature'. The second row of Table 2 is $3\tau_{63}$, the time required for that sensor to show 95% of a step change in temperature (Brock and Richardson, 2001, Chapter 6). A 3 mm diameter PRT in a thermometer screen ventilated at 0.2 m s$^{-1}$, the average level found in this experiment, will take about 195 s to respond to 95% of a change in temperature, five times as long as the 39 s for an identical sensor exposed in an aspirated screen subject to 5 m s$^{-1}$ airflow. The guideline in the World Meteorological

Organization CIMO guide (WMO, 2018) is that a 95% response should happen within 60 s. Without wholly unrealistic assumptions of external wind speed, or some alternative method of increasing airflow across the PRT within the screen, or decreasing the diameter to the PRT to 1.5 mm or less, it would appear unlikely that this WMO guideline response time could ever be attained within a typical Stevenson screen, except in strong winds.

### 240 4.3.4 The specific case of the wet-bulb PRT response time

A wet-bulb thermometer (Twet) is frequently used as part of a psychrometer to derive various humidity parameters, including dew point, and in operation typically comprises a PRT sleeved in a cotton wick, the latter being kept wet by capillary action from an adjacent reservoir of distilled water (see, for example, Meteorological Office 1981, or Harrison 2014 Chapter 6). However, the cotton wick also acts to insulate and dampen the response of the PRT to changes in temperature: informal

experiments suggest that the response time of a dry 'sleeved' PRT at 1.0 m s$^{-1}$ airflow increases by about a factor of three at

room temperature (experimental results ranging from 2.5 to 3.7; see also Table V in Meteorological Office 1981b). (It is non-trivial to design an effective response time experiment with a wetted wick because the resulting temperature response is complicated by evaporation, latent heat, the different specific heat capacities of water and sensor materials, and conduction through the wick.) Rows 3 and 4 in Table 2 show the indicative impact upon response times of an identical sensor sleeved within a cotton wick of the type commonly used for wet bulbs. Following the same logic as above, it can be seen that within a screen ventilated at 0.2 m s$^{-1}$, a typical wet-bulb response time to 95% of a change lies little short of 10 minutes. Even a continuously aspirated wet-bulb – if such a device could be developed to be both practically and operationally feasible – would take almost 2 minutes to attain 95% response for a 3 mm sensor, although an aspirated wet-bulb PRT 1.5 mm in diameter or less could theoretically do so in about 64 s. An Assmann psychrometer (see, for example, Foken 2022, Chapter 8) includes both aspirated dry- and wet-bulb thermometers, but for several reasons (particularly the difficulty in arranging for a constant and reliable supply of water to the wet-bulb wick in all conditions, and in maintaining a clean wet-bulb surface) it is unsuited to continuous automatic operation.

The very different response times of 'bare' and 'sleeved' PRTs illustrate a further issue pertaining to wet-bulb temperatures (and humidity parameters derived therefrom) following rapid changes in air temperatures. Figure 9 illustrates a hypothetical case setting out the cooling curve of two 3 mm PRTs, one a 'bare' dry-bulb (solid line) and one 'dry but sleeved' (dashed line), in response to an instantaneous reduction in dry-bulb temperature from 20 °C to 15 °C, assuming 0.2 m s$^{-1}$ airflow. (This is a deliberately simplified scenario ignoring latent heat effects; the intention here is to demonstrate the impacts of differing response times rather than to suggest a realistic model of a physical wet-bulb sensor.) The $\tau_{63}$ response times of the two sensors are 65 s and 195 s, as given in Table 2, the sleeved sensor being much slower than the unsleeved sensor owing to the insulating properties of the sleeve material. Let us further assume that the relative humidity (%RH) at t = 0 is 85% (for which the wet-bulb temperature at t = 0 would be 18.5 °C), that the ambient %RH remains constant at 85 throughout the change in temperature, and that the response time of the second PRT matches that of an actual wet-bulb. Then it can be seen from Figure 9 that the slower cooling of the 'sleeved' PRT results in a sudden but spurious increase in %RH (dotted line, right-hand vertical axis); from about t = 40 s for over 200 s the calculated %RH exceeds 100 (in which circumstance, by convention, the result is capped at 100%). The true %RH does not return to the nominal unchanged 85 until after about t = 1000 s (about 17 minutes) following the nominal drop in temperature. The dew point (grey dashed line on Figure 9) actually *increases* until t ≈ 30 s, and shortly afterwards follows the dry-bulb curve during the period of nominal %RH = 100.

Events such as outlined above do occur occasionally in the real world, but their transitory nature implies that manual observations of instances of 'wet-bulb thermometer higher than dry bulb' would be infrequent, even at sites where hourly observations were made. Given such circumstances, it would not be surprising if the observer simply ascribed an anomalous 'high' wet-bulb reading to instrumental error, for minor differences – say within 0.2 K, particularly at high %RH – could be expected to lie within instrumental calibration tolerance. The most likely outcome would be that a wet-bulb reading no higher than the dry-bulb would be entered. If the wet-bulb reading *was* entered as higher than the dry-bulb, it is likely that a similar correction would ensue in subsequent dataset quality control processes. However, such events can be expected to be more

obvious where higher temporal resolution is available from automatic weather station datasets, and provided instrumental calibrations are accurately known, they should not necessarily be assumed to be incorrect, but may simply reflect differing sensor response times.

The example of a nominal instantaneous *increase* in air temperatures, less common in meteorological situations, of similar magnitude is illustrated in Figure 10. Here the calculated %RH falls rapidly to below 70 around t = 100 s, and does not recover

to the nominal 85 until about t = 780 s (13 minutes).

The known sources of error when measuring humidity using dry- and wet-bulb psychrometers invite opportunity at this point to refer to the advantages of electronic relative humidity sensors – which include faster response time (except at high %RH), reduced maintenance requirements, lack of dependency upon a water supply, reliable operation below 0 °C, and simplified logger algorithms. For more details the reader is referred to, for example, Burt 2012, Chapter 8, and Harrison 2014, Chapter

6.

### 4.3.5 Implications for sensor averaging time

The WMO CIMO guide (WMO 2018, Annex 1A) recommends a 60 s averaging time for both temperature and humidity sensors. Whilst this is a reasonable statement of requirement, from the above discussion it can be seen that 95% response times

295 ($3\tau_{63}$) of even the fastest commercially-available Pt100 sensors amounts can be expected to be substantially in excess of this when exposed within passively-ventilated Stevenson screens, and longer still when configured as a wet-bulb (i.e. sleeved with a cotton wick). Table 3 illustrates the resulting errors in 1 minute means from 1 … 10 minutes following the previous example in Figure 9 of an instantaneous 5 K fall in temperature (and constant 85 %RH) at t = 0 s, assuming response times given in Table 2. It should be noted that the response times set out in Table 2 are representative of the fastest commercial Pt100 sensors

available in 2020 (Burt & de Podesta 2020), and that the response times of typical Pt100 sensors are almost certainly slower still.

### 4.3.6 Variations of psychrometric coefficient at low airflow

The efficiency of the wet-bulb, and the assumptions used in the psychrometric equation to derive atmospheric humidity parameters from the readings of a dry- and wet-bulb psychrometer, vary significantly with ventilation speed. A value of %RH can be determined by calculating firstly the vapour pressure *e* (in hPa) from equation 4:

$$(4) \quad e = e_s(T_{wet}) - Ap(T_{dry} - T_{wet})$$

and then calculating %RH from equation 5:

$$(5) \quad \%RH = e/e_s(T_{dry}) \times 100\%$$

where $A$ is the psychrometer coefficient (x $10^{-3}$ K$^{-1}$), $p$ the atmospheric pressure (in hPa/1000) and $e_s(T)$ the saturation vapour pressure of water at temperature $T_{dry}$ °C.

315 Harrison and Wood (2012) showed that the value of the psychrometric coefficient $A$ increases steeply from ~ 0.7 x $10^{-3}$ K$^{-1}$ in airflow ≥ 1 m s$^{-1}$ to between 0.8 x $10^{-3}$ K$^{-1}$ and 1.2-1.3 x $10^{-3}$ K$^{-1}$ below 0.5 m s$^{-1}$. The key point here is that screen ventilation has been shown to be almost always below 0.5 m s$^{-1}$ (> 99% below 0.5 m s$^{-1}$, Table 1) and thus $A$ lies in a region where variation with ventilation speed is considerable. This is shown graphically in Figure 11, adapted from Figure 6.18 in Harrison (2014) by adding a shaded box showing 5% and 95% percentiles from the screen ventilation results described here.

To illustrate the possible range of uncertainty, Table 4 sets out calculations of %RH and dew point for several combinations of dry-bulb temperatures and wet-bulb depressions with $A$ varying between 0.7 and 1.1 x $10^{-3}$ K$^{-1}$, corresponding approximately to the 5-95% range of observed in-screen airflow velocities. The calculation method for relative humidity and dew point from dry- and wet-bulb temperatures follows that set out in Harrison, 2014, Chapter 6. The values set out in the table are as follows:

$A = 1.1$ (x $10^{-3}$ K$^{-1}$), assuming a ventilation speed close to zero;

$A = 0.95$, the observed 0.2 m s$^{-1}$ mean value of in-screen airflow found in this experiment;

$A = 0.8$, airflow at about 0.5 m s$^{-1}$, such as might be expected in a Stevenson screen of the type tested with U$_{10}$ ≥ 7-8
330 m s$^{-1}$ approximately, from Table 1;

$A = 0.7$, airflow at ≥ 1 m s$^{-1}$, such as could be expected from an aspirated dry- and wet-bulb psychrometer, such as an Assmann psychrometer, and the in-screen ventilation level assumed in ISO 17714.

It should be borne in mind that increased airflow across the sensors would also result in significant improvements in response
time.

Even at moderate values of air temperature and wet-bulb depression, and necessarily assuming zero rate of change of either temperature, differences between the assumption of ≥ 1.0 m s$^{-1}$ airflow ($A = 0.7$ x $10^{-3}$ K$^{-1}$, ISO 17714 assumption or aspirated sensors) and 0.2 m s$^{-1}$ ($A = 0.95$ x $10^{-3}$ K$^{-1}$, mean in-screen airflow found in this experiment) are at least as great as those resulting from 0.1 K calibration uncertainty in either temperature. For example, at Tdry 10 °C and Twet 7.5 °C, the range in
340 %RH applicable to 0.2 m s$^{-1}$ or 1.0 m s$^{-1}$ ventilation is from 64.8% to 70.0%, and dew point from 3.7 to 4.8 °C; a ± 0.1 K change in dry-bulb temperature would vary %RH by only ±2-3% and dew point by ±0.3 K at such temperatures. The range of variation increases sharply with lower airflow, lower temperatures and greater wet-bulb depressions, such that the combination of Tdry 10 °C and Twet 2.5 °C generates an unrealistic %RH below zero and undefined dew point with $A = 1.1$ x $10^{-3}$ K$^{-1}$ (for a near-zero airflow), in contrast to $A = 0.7$ x $10^{-3}$ K$^{-1}$ (for a 1 m s$^{-1}$ airflow) which produces a more realistic %RH = 16 and
345 dew point -14.4 °C.

## 5. Summary and conclusions

Climate science depends upon accurate measurements of air temperature and humidity, and ventilation speeds within Stevenson-type thermometer screens have important implications for the accuracy and reliability of such measurements. In-screen airflow significantly impacts the response times of sensors within the screen, and affects how representative conditions within the screen are of external air temperature and humidity – especially in conditions of strong solar radiation and/or light winds. This experiment has shown that current assumptions of ventilation speeds within Stevenson screens are too optimistic; the ISO 17714 assumption of 1 m s$^{-1}$ airflow (International Organization for Standardization, 2007) was attained for only 0.01% of the three month experimental period. Of course, the results set out here refer only to a single pattern of screen, sited inland within a relatively sheltered temperate latitude wind climate, for a period of little more than three months. However, the consistency of the results, in a representative wind climate and a typical enclosure and exposure, suggests wider applicability. That is not to suggest that every measurement of temperature or humidity made within Stevenson screens should be disregarded, or retrospectively corrected, even if that were feasible. Instead, based upon sound physical and metrological principles, we should define what is meant by 'true' air temperature (and humidity), and then accelerate the development, piloting and deployment of standardised alternative methods and practices to measure these elements with reduced uncertainty. In doing so, we should of course also take care to provide periods of overlap with existing methods to minimise future data homogeneity issues, particularly for long-period sites. The Stevenson screen will likely be with us for some time to come.

### Data files

The complete experimental results are available on Figshare, https://doi.org/10.6084/m9.figshare.19889515.v1. Two worksheet Excel file; the first sheet describes the file format, the second sheet contains the entire hourly dataset.

### Acknowledgements

Thanks go to my University of Reading colleague Giles Harrison for helpful support and suggestions prior to and during the experiment, and to Ian Read from the Technical Services team for assistance with experimental arrangements and setup. I am also very grateful for helpful and constructive comments from referees which improved this paper.

### Competing interests

The author declares no conflict of interests.

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

**Table 1.** Hourly scalar mean wind speeds at 2 m and within the Stevenson screen for 1 m s$^{-1}$ bins of 10 m hourly scalar mean wind speeds during the analysis period, 13 February to 27 May 2020, University of Reading site. Screen airflow as % external wind speed at 2 m and 10 m exclude occasions when external wind speeds were logged as zero, to avoid divide-by-zero errors. Note that the summary values include all datapoints – see text for differentiated < 1 m s$^{-1}$ and ≥ 1 m s$^{-1}$ subsets, with reasoning.

| Hourly mean wind speed at 10 m $U_{10}$ | Scalar mean 2 m wind speed $U_2$ | Scalar mean 10 m wind speed $U_{10}$ | Mean screen ventilation | Screen % $U_2$ | Screen % $U_{10}$ | Samples | Cumulative samples % |
|---|---|---|---|---|---|---|---|
| m s$^{-1}$ | | m s$^{-1}$ | m s$^{-1}$ | % | % | | |
| 0.01-0.50 | 0.07 | 0.23 | 0.03 | 43 | 14 | 112 | 4.6 |
| 0.51-1.50 | 0.50 | 0.99 | 0.07 | 14 | 7 | 475 | 24.2 |
| 1.51-2.50 | 1.34 | 1.95 | 0.15 | 11 | 8 | 580 | 48.2 |
| 2.51-3.50 | 2.05 | 2.89 | 0.21 | 10 | 7 | 481 | 68.0 |
| 3.51-4.50 | 2.88 | 3.93 | 0.29 | 10 | 7 | 346 | 82.3 |
| 4.51-5.50 | 3.63 | 4.90 | 0.36 | 10 | 7 | 235 | 92.0 |
| 5.51-6.50 | 4.25 | 5.92 | 0.41 | 10 | 7 | 124 | 97.1 |
| 6.51-7.50 | 4.82 | 6.87 | 0.47 | 10 | 7 | 50 | 99.2 |
| 7.51-8.50 | 5.68 | 7.89 | 0.52 | 9 | 7 | 10 | 99.6 |
| 8.51-9.50 | 6.52 | 9.00 | 0.58 | 9 | 6 | 9 | 100.0 |
| >9.51 | 6.78 | 9.60 | 0.60 | 9 | 6 | 1 | 100.0 |
| | | | | | | | |
| **Mean** | **1.96** | **2.80** | **0.20** | **14** | **8** | **2423** | |
| **Median** | **1.75** | **2.50** | **0.18** | **11** | **7** | | |
| *Lower 5%* | *0.06* | *0.50* | *0.03* | *8* | *4* | | |
| *Lower 25%* | *0.94* | *1.50* | *0.11* | *9* | *6* | | |
| *Upper 25%* | *2.83* | *3.90* | *0.28* | *13* | *9* | | |
| *Upper 5%* | *4.39* | *6.00* | *0.45* | *31* | *12* | | |
| *Maximum* | *6.9* | *9.6* | *0.78* | | | | |

**Table 2.** Comparison of calculated response times (s) for a nominal 'bare' PRT of 3 mm diameter in varying airflow, and for the same sensor sleeved in a (dry) wick – here representing two identical sensors where the response time differences are due solely to the presence of the wick around the sensor. Based upon experimental work and empirical relationships described in Burt & de Podesta (2020)

| | 0.2 m s⁻¹ | 1.0 m s⁻¹ | 5.0 m s⁻¹ |
|---|---|---|---|
| $\tau_{63}$ 'bare' PRT | 65 | 29 | 13 |
| $3\tau_{63}$ 'bare' PRT | 195 | 87 | 39 |
| | | | |
| $\tau_{63}$ 'sleeved' PRT | 195 | 87 | 39 |
| $3\tau_{63}$ 'sleeved' PRT | 586 | 262 | 117 |

**Table 3.** Tabulated 60 s running mean values for each minute (the average of the preceding 6 x 10 s sampled values) following an instantaneous reduction in dry-bulb temperature from 20 °C to 15 °C at t = 0, and a nominal wet-bulb similarly, assuming response times for a dry- and wet-bulb thermometer as Table 2. All values given to one decimal place. Note that the nominal wet-bulb temperature is above the dry-bulb temperature until after t = 4 minutes.

| **Time since instantaneous fall (minutes)** | | | | | | | | | | | |
|---|---|---|---|---|---|---|---|---|---|---|---|
| | 0 | 1 | 2 | 3 | 4 | 5 | 6 | 7 | 8 | 9 | 10 |
| *Dry-bulb temperature °C* | | | | | | | | | | | |
| Actual | 20.0 | 15.0 | 15.0 | 15.0 | 15.0 | 15.0 | 15.0 | 15.0 | 15.0 | 15.0 | 15.0 |
| 60 s mean | 20.0 | 18.0 | 16.2 | 15.5 | 15.2 | 15.1 | 15.0 | 15.0 | 15.0 | 15.0 | 15.0 |
| Error K | 0 | +3.0 | +1.2 | +0.5 | +0.2 | +0.1 | +0.0 | +0.0 | 0 | 0 | 0 |
| *(Nominal) wet-bulb temperature °C* | | | | | | | | | | | |
| Actual | 18.5 | 13.7 | 13.7 | 13.7 | 13.7 | 13.7 | 13.7 | 13.7 | 13.7 | 13.7 | 13.7 |
| 60 s mean | 18.5 | 17.9 | 16.7 | 15.9 | 15.3 | 14.9 | 14.6 | 14.4 | 14.2 | 14.1 | 14.0 |
| Error K | 0 | +4.2 | +3.0 | +2.2 | +1.6 | +1.2 | +0.9 | +0.6 | +0.5 | +0.4 | +0.3 |

**Table 4.** Values of %RH and dew point (°C), and the range in values, calculated for various values of the psychrometric coefficient $A$ (x $10^{-3}$ K$^{-1}$) for a selection of values of dry-bulb temperature (Tdry, °C) and wet-bulb depression (K). Parameter combinations producing %RH below zero are indicated as < 0, for which the dew point is undefined (x).

| Tdry °C | Wet-bulb depr., K | %RH at this value of A x $10^{-3}$ K$^{-1}$ | | | | Range in %RH | Dew point (°C) at this value of A x $10^{-3}$ K$^{-1}$ | | | | Tdew range K |
|---|---|---|---|---|---|---|---|---|---|---|---|
| | | 1.1 | 0.95 | 0.8 | 0.7 | | 1.1 | 0.95 | 0.8 | 0.7 | |
| 30 | 2.5 | 79.9 | 80.8 | 81.7 | 82.3 | 2.4 | 26.2 | 26.3 | 26.5 | 26.6 | 0.4 |
| | 5.0 | 61.5 | 63.3 | 62.1 | 66.3 | 4.8 | 21.8 | 22.3 | 22.7 | 23.0 | 1.2 |
| | 10.0 | 28.8 | 32.4 | 36.0 | 38.3 | 9.5 | 9.9 | 11.7 | 13.3 | 14.3 | 4.4 |
| 20 | 2.5 | 73.6 | 75.2 | 76.9 | 77.9 | 4.3 | 15.1 | 15.5 | 15.8 | 16.0 | 0.9 |
| | 5.0 | 49.1 | 52.3 | 55.6 | 57.7 | 8.7 | 9.0 | 9.9 | 10.8 | 11.4 | 2.4 |
| | 10.0 | 4.8 | 11.3 | 17.8 | 22.2 | 17.3 | -21.1 | -10.9 | -5.1 | -2.2 | 18.9 |
| 10 | 2.5 | 61.7 | 64.8 | 67.9 | 70.0 | 8.3 | 3.0 | 3.7 | 4.4 | 4.8 | 1.8 |
| | 5.0 | 25.7 | 31.9 | 38.1 | 42.2 | 16.5 | -8.8 | -6.0 | -3.6 | -2.2 | 6.6 |
| | 7.5 | < 0 | 0.8 | 10.1 | 16.3 | x | x | -45.9 | -20.1 | -14.4 | x |
| 0 | 2.5 | 35.6 | 41.8 | 48.1 | 52.2 | 16.6 | -13.4 | -11.4 | -9.7 | -8.6 | 4.8 |
| | 5.0 | < 0 | < 0 | < 0 | 7.7 | x | x | x | x | -30.7 | x |

# Figures

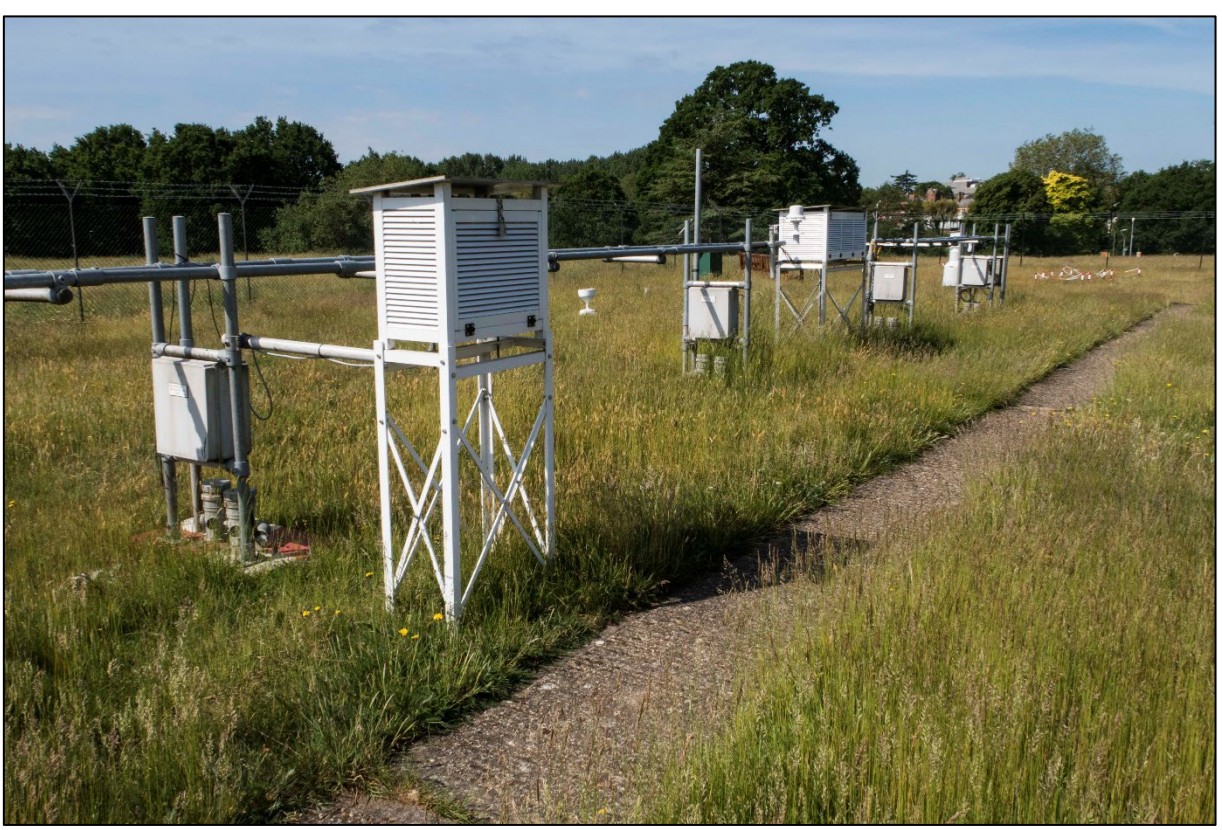

**Figure 1. Within the Reading University Atmospheric Observatory enclosure: the screen nearest to the camera housed the Gill Windsonic anemometer used in this experiment, and its screen door (shown with padlock) opens to true North. This photograph was taken on 27 May 2020. Owing to the coronavirus emergency legislation, the university campus had been closed for two months at this time and normal grass cutting and maintenance work suspended. Photograph © Copyright Stephen Burt**

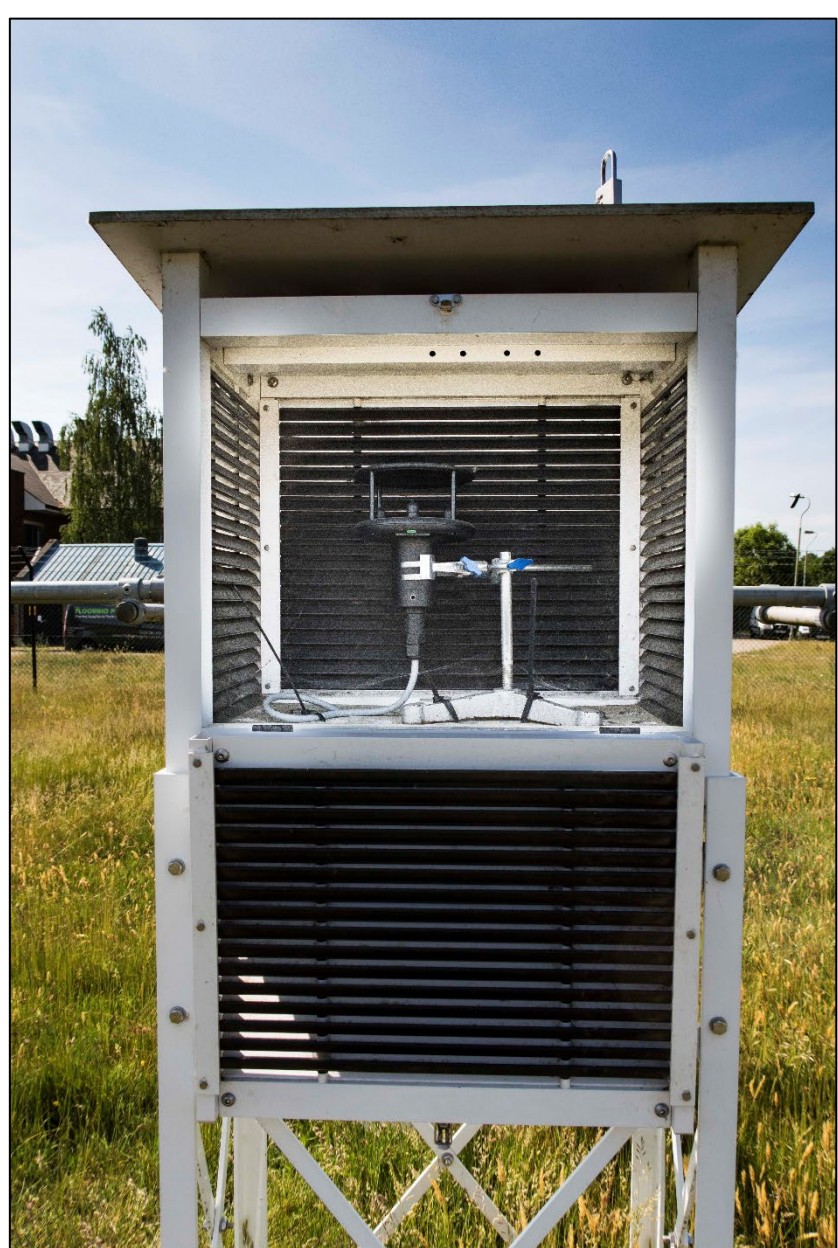

**Figure 2. The Gill Windsonic anemometer located within the Stevenson screen shown in Figure 1. The inside dimensions of the screen were 50 cm width x 25 cm depth x 43 cm height. This photograph was taken on 27 May 2020. Photograph © Copyright Stephen Burt**

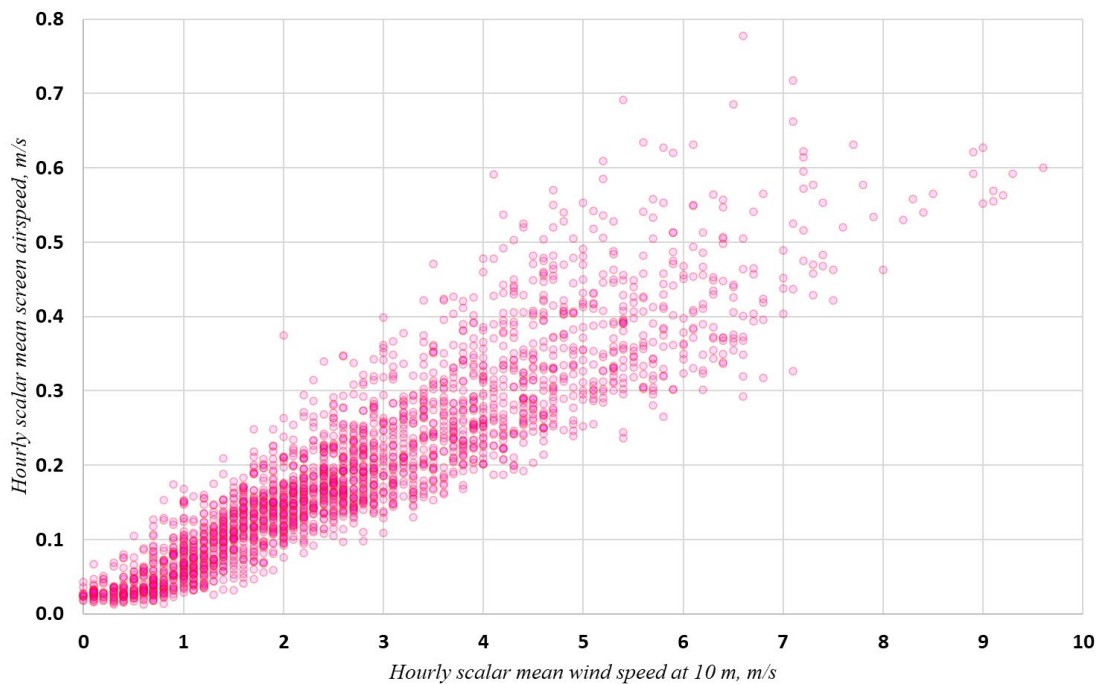

**Figure 3. Hourly scalar mean wind speeds within the screen ($U_{screen}$) plotted against the external 10 m scalar mean wind speed $U_{10}$, for the period 13 February to 27 May 2020. Units m s$^{-1}$.**

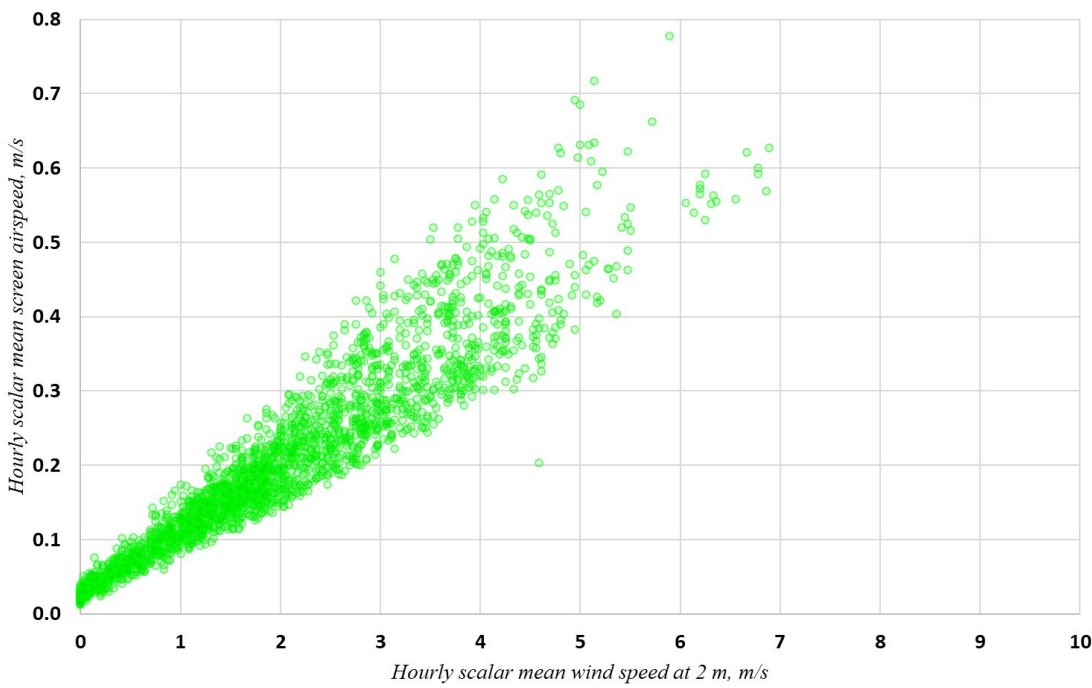

**Figure 4. As Figure 3, but for 2 m scalar mean wind speed $U_2$.**

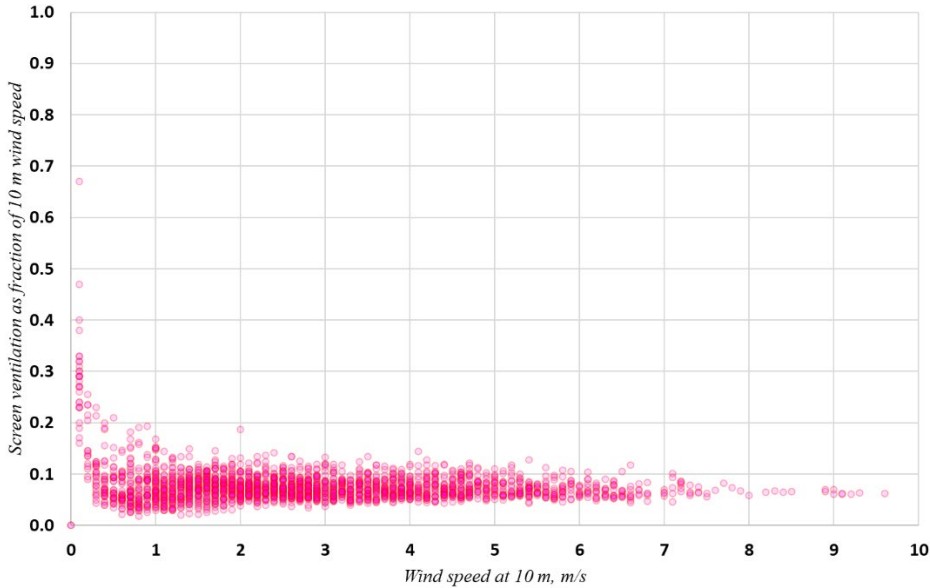

**Figure 5. Scatterplot of $U_{screen}$ as a fraction of $U_{10}$, based on hourly scalar means. Above about 1 m s$^{-1}$ $U_{screen} \approx 7\%$ of $U_{10}$**

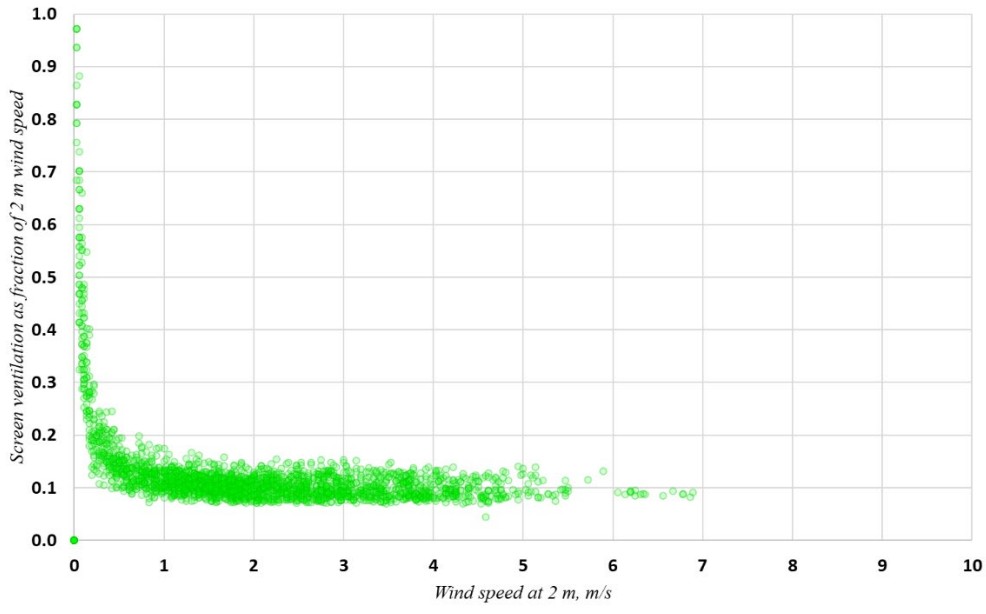

**Figure 6.** As Figure 5, but for $U_2$. A few values of $U_{screen} > U_2$ are omitted for clarity – see text.
Above about 1 m s$^{-1}$ $U_{screen} \approx 10\%$ of $U_{10}$.

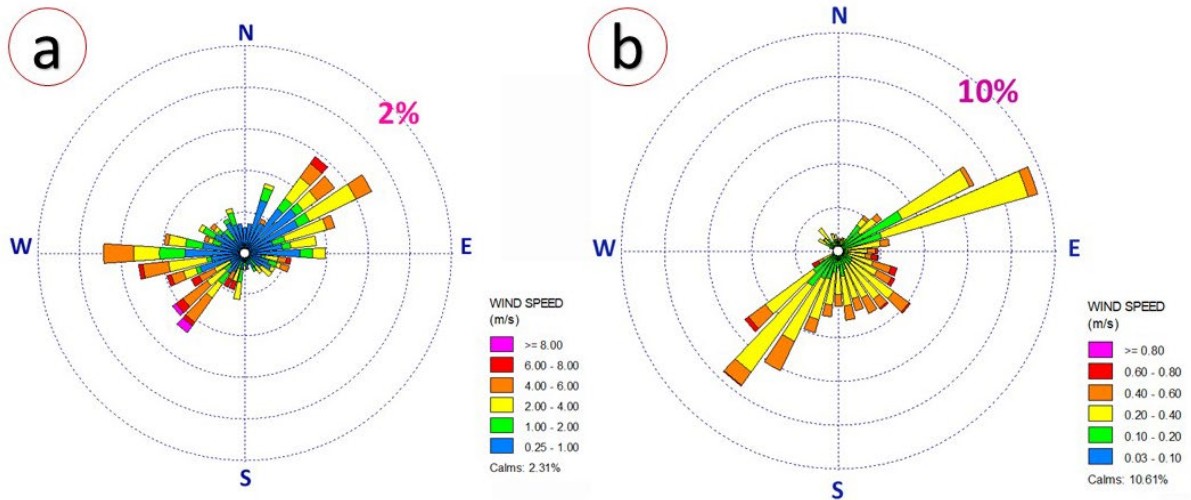

**Figure 7. Hourly vector mean wind direction frequencies, in wind rose format, during the experimental period: a, at 10 m; b, within the Stevenson screen. Note that the scales and class boundaries necessarily differ - the outer scale ring representing 2% frequency for the 10 m plot and 10% for the screen. Mean wind speed 2.8 m s⁻¹ at 10 m, 0.20 m s⁻¹ within the screen.**

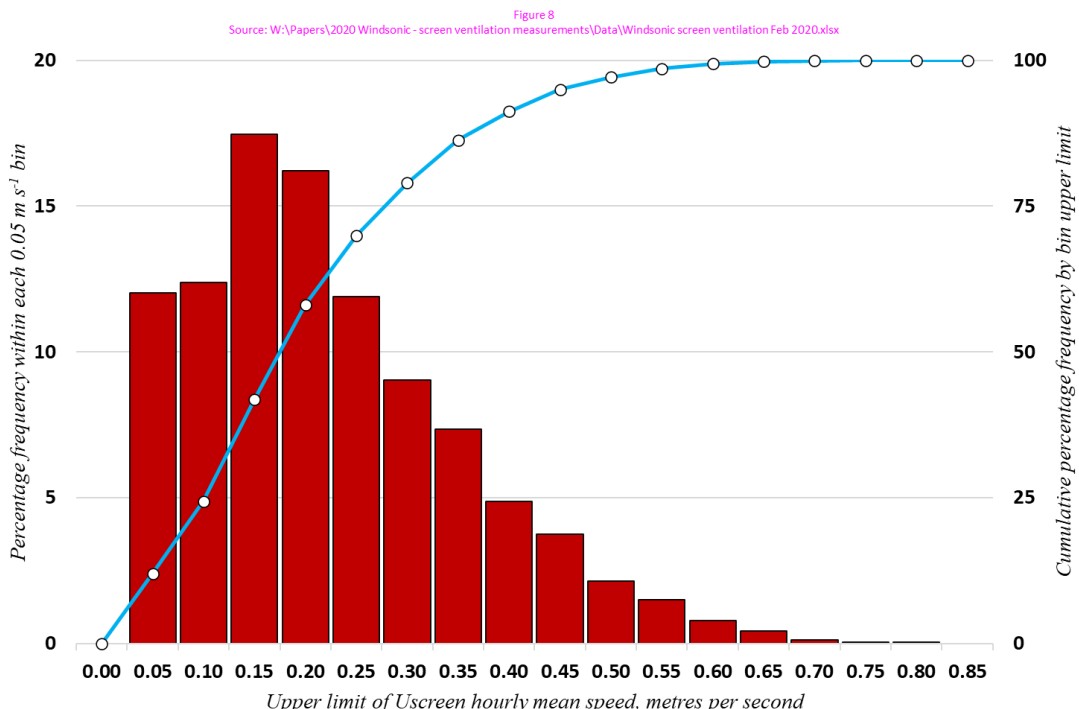

**Figure 8. Percentage frequency distribution of hourly mean screen ventilation U$_{screen}$ within 0.05 m s⁻¹ bins. Red columns show per-bin frequency (left vertical axis), blue line and markers cumulative percentage frequency below upper bin limit (right vertical axis). Total 2423 observations, hourly mean speed 0.20 m s⁻¹, median 0.18 m s⁻¹, minimum 0.01 m s⁻¹, maximum 0.78 m s⁻¹.**

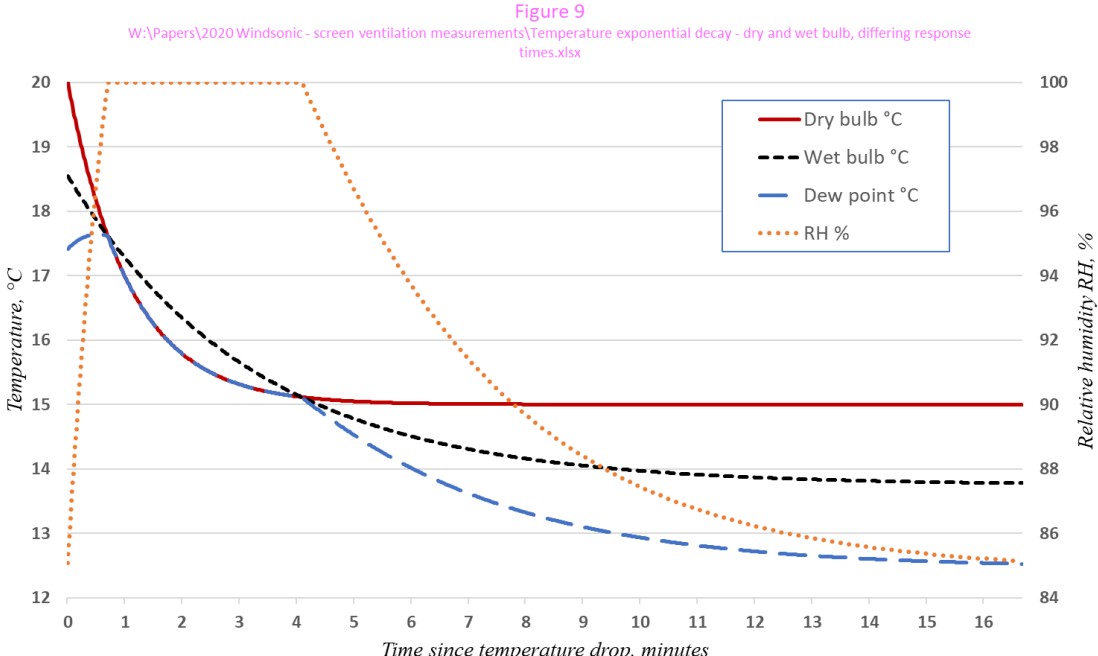

**Figure 9. Time series (minutes) of dry-bulb, nominal wet-bulb and calculated dew point temperatures (°C, left axis) and relative humidity (%RH, right axis) following an instantaneous *fall* in temperature of 5 K from 20 °C, assuming sensor response times per Table 2**

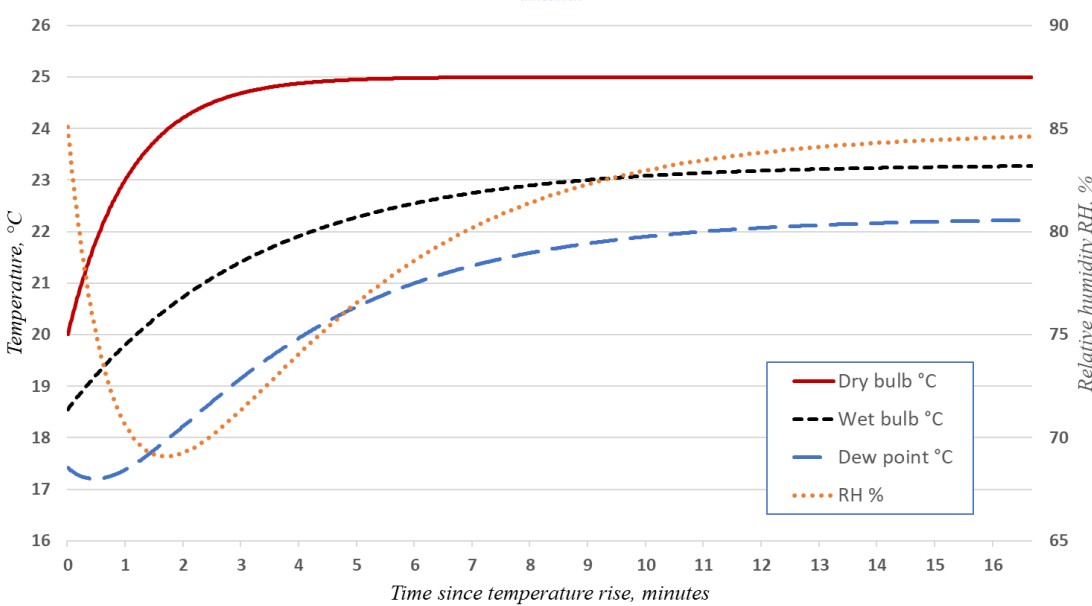

515

**Figure 10. As Figure 9, but for an instantaneous *rise* in temperature of 5 K from 20 °C**

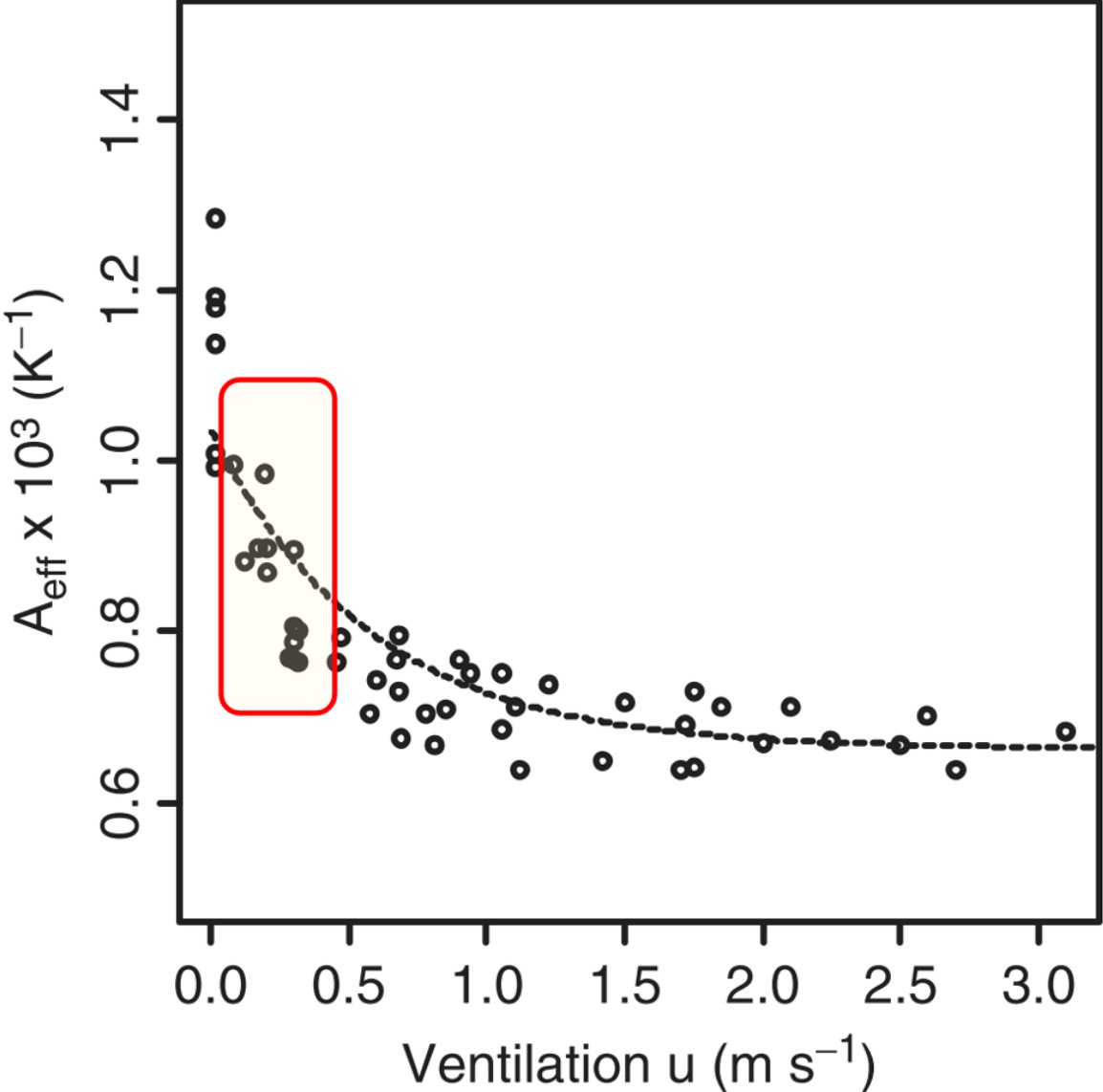

**Figure 11. Dependence of psychrometer coefficient *A* on ventilation speed. From Harrison (2014), Figure 6.18. Shaded box overlay shows the 5 and 95 percentile limits of screen airflow observed during this experiment.**

520