# Peer review of "Measurements of natural airflow within a Stevenson screen, and its influence on air temperature and humidity records"

_Geoscientific Instrumentation, Methods and Data Systems, 2021_

## Author Response (AR1)

**Referee comments for Airflow within Stevenson screen, and responses – February 2022**

Mike Molyneux

General comments

The paper is directly relevant to many locations where timeseries of temperature measurements are made around the world.  The work examines an underpinning assumption of one of the common methods of air temperature measurement.  The conclusion is useful, and helps add to both improvements to these timeseries in future and to assesing the uncertainty of measurement in the past.  The method and assumptions are clear so that the results can be highlighted robustly.  There is a good list of relevant references and credits and the papers title accurately reflects its content and it has a suitable abstract.  The presentation is clear and language used is appropriate.  The formulae are correct and no parts of the paper need to be removed.

Specific comments

Figure 8 shows an interesting case but might benefit from some discussion of manual observing practices and/or quality control.  In the case of having a person observe the readings (now or in the past) they will take special care if the wet and dry bulb values are close to each other and will be suspicious when the wet bulb reading is above the dry.  There will have been  many occasions when a sudden temperature drop could have occurred and yet I know of no special processes for treatment of errors as large as the maximum case shown. Anecdotally this suggests it wasn't a common problem, although it may be related to the opening of the door required for a human to read the values.

Discussion of anemometer performance approx line 143

The anemometer will measure the resolution quoted by the manufacturer, but very low wind speed performance may be assumed rather than tested.  While this is unlikely to have a significant impact it could be discussed for completeness.

Technical corrections

Non noted

My responses – posted online

I thank Mike for his helpful comments on the draft, both in general and the two specific points made -

The first, regarding the lag of the wet bulb behind the dry bulb in rapidly changing temperatures, relates to my Figure 8, a hypothetical comparison of two sensors where the response time of one (the wet bulb) is much slower than the other (the dry bulb). For simplicity only response time differences were considered, deliberately disregarding latent heat changes and conduction through the wet bulb muslin. The effects suggested in my Figure 8 are not uncommon although they are of course transient, and tend only to be noticed in terms of Twet > Tdry when the humidity is high (Twet close to Tdry even before the step change) and when the change is sufficiently rapid and of sufficient magnitude. Not surprisingly, close examination of short-period logger data (1 min or less) reveals more instances than are evident from, for example, daily or manual hourly observations by a human observer. Even where Twet does not exceed Tdry, an increase in humidity (RH, where calculated from dry- and wet-bulb readings) relative to, say, an adjacent capacitance sensor can easily be ascribed to the drop in temperature. Careful comparisons of RH measurements from adjacent sensors (capacitance sensor against Tdry/Twet RH) during abrupt changes in temperature often show a short-term relative increase in the latter, although of course capacitance sensors are inherently less responsive anyway at high RH. But such relative differences can be found in the observational record. In any case, it is easy to explain minor differences - say to +/- 0.2 K - as being within instrumental calibration tolerance, and thereby disregarded.

With regard to Mike's second point, relative accuracy of low-speed airflow measurements, the comment is fair and I will happily include this in the revised paper if it is duly accepted for publication.

Stephen Burt, University of Reading

10 January 2022

Referee 2 (anonymous)

The manuscript was interesting and relevant to better understand the sampling characteristics as function of external wind flow in a Stevenson screen shelter. Many Met Services and other agencies still use the Stevenson Screen as their primary framework for taking measurements of temperature and humidity. Results from studies like this could help understand measurement uncertainites and also provide wind dependent correction factors for measurements using Stevenson screen.

Overall, I think the paper was well written and provided new results that will be interesting for readers. However, I feel the paper was limited in scope and could be expanded to better understand flow characteristics in the Stevenson screen at different locations within the screen. The could further be expanded to collect actual measurements of temperature and pressure to compare the variability of measurements with observed environmental wind speed and direction observations at the height of the Stevenson screen. I would like to see the author to expand on these ideas for future studies.

> I thank the reviewer for the suggestions. With respect however, this paper was never intended as a definitive statement of the variation of airflow *within* the screen, and its title reflects that. Indeed, understanding 'flow characteristics in the Stevenson screen at different locations within the screen' in any detail would suggest a combination of (small) multi-sensor and CFD approach rather than a programme of exploratory measurements with a single sensor as was the purpose of this experiment. Some work regarding a CFD approach has been attempted, and is referenced in my paper (Dobre, et al). The work documented was, and is, deliberately limited in scope to quantify the range of typical ventilation speeds occurring within a Stevenson screen, measurements hitherto lacking although often assumed in various important areas such as psychrometric coefficient and response times, as my paper points out. Observed ventilation rates are then compared with conventional measurements of wind speed at standard heights, in order to provide guidelines for the wider meteorological community of how standard wind measurements can be used to infer in-screen ventilation rates, and onwards to suggest occasions when responsive and accurate measurements of air temperature within thermometer screens of the Stevenson type may be less reliable.

> A separate but related project is ongoing to document observed differences between aspirated and Stevenson screen measurements of air temperature, for which these results will be directly relevant. Some results have already been published (Harrison, R. G. and S. D. Burt, 2021: Quantifying uncertainties in climate data: measurement limitations of naturally ventilated thermometer screens. Environmental Research Communications, 3, 061005).

Below is some specific comments for consideration:

Lines 55-62: The wind sensor was mounted in the center of the screen.  Was this representative of the location of where temperature and humidity measurements are typically made?  If not, why not mount the wind sensor at that location(s)?  Was there any thought of makin flow measurements at other locations (higher/lower, closer to the screen walls, etc.) to see to characterize the variability in the screen.  Significant variability could impact the observations of the temperature/humidity measurements.  Did you explore any impacts of the measurements while using the laboratory stand to mount the sensor?

➢ The siting of the sensor was intended as far to match the typical location of temperature and humidity sensors within this type of enclosure. Agreed that it would be interesting to understand the variability of the flow at other points in the screen, but this would present experimental difficulties with the current apparatus mainly owing to the size of the sensor – there was simply not enough room to fit two such units within the screen (more than one unit would in any case complicate airflow within the screen). To determine variations within the screen structure, an experimental design could be envisaged using multiple small hot-wire anemometers (and probably within a wind tunnel), and the results used to develop a CFD model, but as stated above variation *within* the screen was not the primary motivation of the experiment as described.

Lines 63-64: Were the wind measurements logged at 1 min, 5 min, and hourly or was the observations logged at 1-min and averaged to 5 min and hourly or were subsampled at 5 min and hourly?  This is a bit confusing.

➢ The sensor was sampled at 1 Hz and logged at 1 min, 5 min *and* hourly. Logged samples included average, minimum and maximum speeds, and vector mean directions. For most of the analysis, hourly means were sufficient, although 1 min and 5 min records were available and were examined where additional detail was beneficial. There was little point in providing additional analyses based upon the 1 min and 5 min records when conclusions using these data differed little from that derived using hourly values.

In Fig. 1, it would be interesting to know what is the direction of North for reference.  This could help understand if there were any impacts of flow if the environmental wind was directly along one of the corners for example.

➢ Added note on orientation to Fig 1. In the northern hemisphere midlatitudes, screen doors open to the north and I had assumed this was common knowledge. The possible impact of the corners of the screen structure was examined in section 4.2.

Lines 128-130: The external wind speeds are measured at 2 m and 10 m.  What is the height of the wind sensor above ground inside the screen?  If the sensor in the wind screen is not at 2 m, what is the potential impact in the results of the study?

➢ The sensor within the screen was located at 1.25 m above ground level, the standard height within the UK of temperature and humidity sensors when exposed within a Stevenson screen. While wind speeds at this level are available from the observatory records, it is not a standard height for wind records and thus would make the results less relevant to other sites with wind records at standard heights (2m and/or 10 m).

I found the results shown in the discussion sections 4.3.3-4.3.6 interesting and a nice exercise to explore the potential impacts.  What would make this paper (or future paper) even more interesting if these results could be verified with actual observations from a Stevenson screen comparison study.

➢ Again with respect, I fear the referee has misunderstood the purpose of this particular research, which is not to undertake comparisons between different Stevenson screens – although we are in fact accumulating data towards something similar, and the results will be published in due course. It would lengthen and dilute the current paper unduly to include these comparisons.

Editor – 18 Jan 2022

Dear Stephen Burt,

Based on the overall positive and constructive nature of the responses, I invite you to reply to the remaining referee comments and, based on this feedback and anything else that you may wish to improve, submit a revised manuscript.

All my best,

Andy

Referee 3 (Stephanie Bell)

Stephanie Bell, NPL, 6 Feb 2022

Overall

This is an interesting and important piece of work. I have made some comments intended to help clarity and impact for the readership.

As I read the manuscript, I wondered whether this work would be better presented as two papers: one focusing on the experimental work, and one discussing the influences of airflow on temperature error and of temperature error on wet-bulb temperature (i.e. mostly section 4.3)?

➢ It did in fact start off that way, but in preparing the material it appeared that presenting the first topic on its own would be likely to invite the question 'Tell me why this is important?', while similarly the second question could be parried by 'But when and why would this occur'?, and therefore that such a division would be incomplete.

Title

I am not sure the title as it stands is enough to convey the full significance of the work. If the manuscript remains one item, the existing title "Measurements of natural airflow within a Stevenson screen" might usefully continue ", and the impact on airflow-sensitive measurements of temperature" or something similar. (This also points to how the manuscript might be divided.)

➢ A good suggestion. I have revised the paper's title along the lines suggested, but I would prefer not to divide it for the reasons given above.

Abstract

This is clear, and it conveys the broader context and implications of the measured values of airflow.

Manuscript details

Line 58: Although the text says the anemometer  was visually centred in the Stevenson screen, it appears visibly off-centre in Figure 2.

➢ In fact the sensor was positioned as close to the centre of the screen volume as possible, but I agree Fig 2 makes it appear slightly off-centre. The difference is only a few centimetres, however. I have amended the text appropriately.

Line 64: It is good to see the statement about calibration. However, it would be good to know how current that calibration was. If not recent, it would also to desirable to note what the expected level of calibration drift might have been (if any) for this anemometer type. Was that uncertainty 2 % of reading or a fixed uncertainty of 2 % of full scale? Was there any lower limit of range where the uncertainty rose above 2 %? In addition, it would be desirable to give the coverage probability and coverage factor for the 2 %. Finally, it would be good to know whether the uncertainty in using the anemometer is predominantly only that of calibration, or somewhat larger, as is the case in many types of measurement. Overall, the resulting uncertainty in the rather small windspeeds measured would depend on these things.

> Reliable and accurate calibration is always important – of course – but calibration uncertainty simply isn't the main factor here. The conclusions set out in the paper are insensitive to even fairly large uncertainties in the sensor's low speed calibration: even if the calibration at 0.2 m s$^{-1}$ (the mean in-screen speed logged during this experiment) were out by +20%, which is 10x manufacturer spec, this would change the ratio of interior:exterior wind speed only slightly (from 10% to 12% for 2 m, and similarly from 7% to 9% for 10 m). While more uncertainty attaches to the lowest speeds, this is largely irrelevant to the outcome as the stopping speed of the external Vector anemometers meant that reliable comparisons below U2 or U10 < 0.4-0.5 m s$^{-1}$ could not be obtained in any case. But the point is a fair one, and accordingly I have added an extra note in the paper to set out details of the calibration of both sets of instruments.

Line 135: among the reasons for selecting the cup anemometer, was it also because they were available and maintained?

> Yes

Was there, or would there an opportunity to compare the two anemometers directly at relevant airspeeds, as a confirmation of consistency between the two?

> The 2 m and 10 m anemometers in the observatory are operational instruments and could not be easily removed for comparison without disrupting other programmes. However, the Sonic anemometer in use here had previously been compared side-by-side with an identical pattern of Vector Instruments anemometer of known calibration over a 4 week period for exactly this purpose, and again afterwards for several months, and the two instruments agreed within 2% over a wide range of observed speeds, except at low speeds owing to the 0.3 m s$^{-1}$ stopping speed of the  Vector anemometer.

Line 158: U2 (at 2m height) is "not shown", but this feels a little disappointing, given that a relationship for this is derived at line 165. (Also, should these questions be numbered and referred to from the text?)

➢ The scatterplot for U2 speeds was prepared but not included as it seemed unnecessary. However, at the referee's suggestion I have included in the same format as Fig. 3.

Figure 4 caption appears to be missing.

➢ This appears to be a glitch in the publisher's PDF creation as the caption is included in my MS. I will check it appears in the updated file.

For the graphs in Figs 4, 5 and7, the title above the graph can be removed.

➢ Agreed, these are for my reference only and would be annotated for removal at proof stage if not before.

Line 187: where the text says "lower tha[n] for winds □□1 m/s" does this really mean "lower than then the data would suggest"?

➢ I have reworded to ' … the ratio of Uscreen to U2 and U10 for wind speeds < 1 m s-1 is probably little different to that for winds ≥ 1 m s-1.'

Line 207: all observations or all means?

➢ I have reworded to '… all 2423 hourly means'.

Fig 7: It is a little hard to see how the percentages of winds relate to the values in table 1, especially for values in the range 0 to 0.05 m/s.

➢ Perhaps I misunderstand the point being made here, but Table 1 refers to 10 m wind speed classes (for which the lowest bin is 0-0.5 m s$^{-1}$), whereas Fig 7 relates to in-screen ventilation speeds, with lower bin 0-0.05 m s$^{-1}$.

Is the mode (most common value) different from the mean? This would be relevant to report. (Perhaps consider whether this is relevant to mention in the abstract too?)

➢ All are positively skewed, as would be expected with a distribution bounded by zero:
➢ In-screen ventilation: mean 0.20 m s$^{-1}$ (Table 1), mode bin 0.15-0.20 m s$^{-1}$ (Fig 7), median 0.18 m s$^{-1}$ (from original dataset); distribution also given on Fig. 7 in original paper (now Fig. 8)
➢ U2: mean 1.96 m s$^{-1}$ (Table 1), mode bin 1.51-2.50 m s$^{-1}$ (original data), median 1.75 m s$^{-1}$ (from original dataset)
➢ U10: mean 2.80 m s$^{-1}$ (Table 1), mode bin 1.51-2.50 m s$^{-1}$ (Table 1), median 2.50 m s$^{-1}$ (from original dataset)
➢ I have added median values to Table 1 and Fig 8.

Line 236: what does "preferential orientation of eddies mean" (or would most readers not need that explained)? If all air movement inside the screen is turbulent, does the mean that the anemometer measures "net wind speed" and that eddy windspeeds on a microscale might be greater, i.e. the anemometer does not have fine spatial resolution? If so, might it underestimate the micro-scale windspeeds?

> A definitive answer to this question would require a greater density of high-resolution (> 1 Hz) small sensors operating within a wind tunnel environment, coupled with CFD modelling; it is outside the scope of the paper.

Line 256: References say that warming "occasionally" amounts to 2-3 K, but it would be helpful to mention what level of warming is thought to occur "commonly".

> Half a degree is not uncommon. I have added this comment to the manuscript.

Line 279; "without any cladding" …? Perhaps "uncovered"

> Agreed.

Line 281: It is not clear why 3⎕63 is the time required to achieve 95 % of a step change. Is there a further explanation?

> This follows from response time theory; I have added a reference

Table 2: Perhaps say here, or earlier, what is the relevance of sensor in a dry wick?

> Added a sentence to explain that the response time of the 'sleeved' sensor is compared with an otherwise identical and unsleeved sensor in the same environment – i.e. the difference in response time is down to the insulating effects of the wick/sleeve.

Line 304: "an aspirated wet bulb – if such a device could be if such a device could be developed …" these exist and are in widespread use - for example Assmann psychrometers and many others, and even a historic design by the WMO.

> Agreed, but an Assmann psychrometer is not suitable for continuous automatic use. The issue lies not with sensors or methods of ventilation, but entirely in maintaining a constant and reliable supply of water to the wick in all circumstances (high and low humidity, and in particular temperatures below freezing, and maintaining a clean wick).

The flaws of wet-bulb measurement leave an opportunity to mention the advantages of electronic relative humidity sensors. It seems rather an omission not to.

> Agreed.

The term "wicked" is problematic as it is open to reading with another meaning (as in "wicked witch"). Once seen, this is hard to unsee and could distract readers. The WMO No8 CIMO Guide avoids this word, in favour of other terms such as wet bulb, wet-bulb sleeve, and similar.

> Ha ha! Agreed. I hadn't read it that way, but now I can't 'unsee' the connection. Amended.

Many other sources of error affect wet- and dry-bulb hygrometers – it seems an omission not to mention them, and their magnitudes, for perspective.

> It is easy to dilute the focus of the paper by delving into other issues, but I will include a short note to this effect.

Fig 8: it is not completely clear whether these are all calculated values, or not. A bit hard to follow – maybe start this description with an overview to orient the reader?

> These are all calculated values. I will reword to clarify as suggested.

Line 336 a constant relative humidity during a 5 K fall in temperature seems slightly unlikely, and this distracts slightly from the point being made.

> Agreed to a point, but at 1 K the differences are of course less obvious. It is a theoretical construct to show the point being made, but in temperate latitudes falls in temperature of 5 K in a few minutes are not uncommon, and are not uncommonly accompanied by *falls* in relative humidity (dew point falling faster than air temperature) – particularly at sharp frontal passages or in thunderstorm downdraught situations.

The "unit symbol" for "percent relative humidity" is weakly standardised and it is accepted to use "%" or "%rh" is also widely used. In either case, there is a space between the number and the symbol.

> I have compromised and am happy to use %RH; the paper has been amended accordingly with definition at first usage

340 Is the "spot mean" a rolling mean (of 6 values here?). Is "spot" a recognised term?

> 'Spot' in datalogger terms is usually taken to mean 'near-instantaneous sample', but I have amended Table 3 wording to make this clearer.'

4.3.6 What about mentioning the accepted published values of the psychrometer coefficient (still air and aspirated with moving air ⎕1 m/s) for context? A key point that the case in point is between these regimes. The values by Harrison and wood remain of interest of course and there would be scope for further study of this.

\>        Accepted values of the psychrometer coefficient are given in 4.3.6 and in Fig. 10 (from Harrison and Wood); I'm not sure I understand what other values are being suggested?

Table 4: What does X designate?

➢ Where the calculation using the parameters given in Table 4 generates an unrealistic RH (i.e. below 0%, for which dew point is not defined). This should have been made clearer in the table caption – now amended to do so

381 A=1.1 is very far from the accepted published value for still air.

➢ I would question whether there really is an accepted value of A for still air. However, 1.1 is derived from Harrison & Wood (my Fig 10), while Harrison (2014), Chapter 6, Fig 6.18 suggests values for 0 m s$^{-1}$ between 1.0 and 1.3. Fig 10 (now Fig 11) has been updated to reflect this. I'm not I quite follow the referee's point here.

---

## Author Response (AR2)

**Copernicus comments – 26 May 2022**

Comments to the author:
Dear Dr. Burt,

Following the positive reviewer comments, I have taken a pass through the parts of the manuscript that they have highlighted and have made a few additional notes myself. These comments are largely clarifications, requests for more information on your methods, and related to house style. The one request that departs from this "minor" approach is that regarding our data-availability policy, which also extends to model code.

I look forward to reviewing a revised draft with these additional data components.

Best wishes,

Andy Wickert

SPECIFIC COMMENTS:

Dr. Molyneux commented on Figure 8, and you left an extensive response to this, but without changing the manuscript text. If such a long response be warranted, then I would imagine that an update to the caption as Dr. Molyneux requested may also be appropriate.

A substantive comment/response on the point raised was included in my 24 February revision (tracked version, lines 307-318) and is retained in the latest version – section 4.3.4, paragraph commencing 'Events such as outlined above do occur occasionally in the real world …'

Regarding Dr. Bell's comments:

Line 58 (original submission): Could you amend the text to indicate the amount by which the anemometer might be off-center, as you have noted in the response?

This was already amended in the 24 February revision to read as 'close to the centre'. The exact dimensions are not material to the argument, and were not recorded: the experimental setup was dismantled at the close of the field experiment. Figure 2 makes the position of the wind sensor sufficiently clear.

Line 64 (original submission) and lines 219-224 (your tracked changes version). This information seems misleading because the external sensor is out of spec at this point. So therefore, no reliable comparison could be made and concerns about the internal sensor's error don't seem so meaningful without considering the external wind sensor as well.

With respect, I believe I fully covered this point previously – 24 February tracked version, section 3.1.2, lines 219-224, where it is unambiguously stated that 'While it is possible that more uncertainty attaches to the lowest speeds, this is largely irrelevant to the outcome, because the typical 0.4-0.5 m s-1 stopping speed of the external U2 and U10 Vector anemometers meant that reliable comparison ratios could not be accurately obtained below these levels.' I believe it is preferable to include the lower wind speed values with a rider to the effect that the sensitivity of the external sensors was insufficient fully to detail the performance in this area, and leave it up to the reader to decide whether or not to put any faith in this part of the analysis, rather than simply omit it without further comment.

Line 381 (original submission): I think that the problem is that the table is not clear that the psychrometric coefficient here is multiplied by 1E-3. Adding a "x" symbol to the caption would help, and I think that something in the table itself would too.

Done. The factor for A, x $10^{-3}$ $K^{-1}$, has been added to the table caption and the table heading. For clarity in the heading this has necessitated a smaller font size in order that it be included in one line - personally I think the latter is unnecessary as the factor is explicitly stated in the caption. The factor is now explicitly stated everywhere the value of A is referenced within the accompanying text, eight times in all.

My comments:

General: Do you have a reference to demonstrate that most temperature measurements are taken within Stevenson-type thermometer screens?

Included as requested.

General: Please go through the style guide (https://www.geoscientificinstrumentation-methods-and-data-systems.net/submission.html) and bring the manuscript into the house standards. Number equations and avoid footnotes if possible.

Done

General: Please include your data and code (per policy: https://www.geoscientific-instrumentation-methods-and-data-systems.net/policies/data_policy.html) in a linked repository. I typically use GitHub+Zenodo for my own work (with code or small data sets); you may have other preferences.

Uploaded to Figshare, https://doi.org/10.6084/m9.figshare.19889515.v1 , and included in text

Line 107, tracked-changes version: Both instruments do not agree, but rather are offset by a ratio, as you demonstrate.

I believe the wording used correctly and succinctly expresses the relationship.

Section 3.1. It would be good to have goodness-of-fit parameters here, alongside which subset of the data was fit based on the anemometers' performances.

Noted in 3.1 that all 2423 hourly values were used in the analysis, subsequently segmented into the two wind speed classes whose reasoning is explained in detail within the text. Summary statistics related to goodness-of-fit are now included in section 3.1 (for values of U2 or U10 > 1 m s-1) and in Table 1 (for the entire dataset).

OTHER POINTS

Submission notes refer to **Figure S1 to be included in the Supplement** – but there is no Figure S1 in my MS.